# Macrocycle as a “Container” for Dinitramide Salts

**DOI:** 10.3390/ma15196958

**Published:** 2022-10-07

**Authors:** Sergey G. Il’yasov, Vera S. Glukhacheva, Dmitri S. Il’yasov, Egor E. Zhukov

**Affiliations:** Laboratory for High-Energy Compounds Synthesis, Institute for Problems of Chemical and Energetic Technologies, Siberian Branch of the Russian Academy of Sciences (IPCET SB RAS), Biysk 659322, Russia

**Keywords:** tris(carbohydrazide-N,O)nickel (II) dinitramide, dinitramide, complexes, nickel (II)

## Abstract

Dinitramic acid salts are promising components as oxidizers and burning-rate modifiers of high-energy compositions. However, most of these salts are not free of drawbacks such as hygroscopicity. Therefore, their application under special conditions of use and storage is limited. The synthesis and storage of stable dinitramic acid salts is a topical issue. Here, we synthesized an adduct starting from the nickel salt of dinitramic acid with carbohydrazide and glyoxal to settle the problem of stability and storage of that salt. The chemical composition of the adduct was confirmed by infrared spectroscopy and elemental analysis. The Ni content was determined by atomic emission spectroscopy. Thermogravimetric DSC and TGA analyses showed the adduct to have three decomposition stages. The adduct exhibits a good thermal stability and a low sensitivity to mechanical stimuli. Here, the adduct is demonstrated to be a promising burning-rate inhibitor of pyrotechnic compositions.

## 1. Introduction

Oxygen compounds have an important role to play in many high-energy rocket propellant and explosive compositions as oxygen suppliers to regulate the formulation. The oxidizers such as ADN, CL-20, and NH_4_ClO_4_ have been recognized as the most energy-rich components to make high-performance rocket propellants. Ammonium dinitramide (ADN) holds promise as a component of green energy production [1,2,3,4]. Despite dinitramic acid (DNA) salts being abundantly known and well-studied, the need for the synthesis of new compounds is persistent at present as well, which is most notably due to the high content of oxygen per unit weight and environmental safety of both the product itself, its combustion products, and production technology [5,6,7,8,9,10,11,12,13]. However, ADN is not free of some drawbacks that hinder its application as the component of rocket propellants. First, it is a hygroscopic substance that requires compliance with special conditions (dry air, certain steady air moisture, inert gases, protection from static electricity) in the production technology and storage of ADN, as well as items based thereon. Second, it is well-soluble in water and many polar organic solvents, and requires great energy inputs for release from an aqeous solution into the crystalline state. Third, it begins to decompose at a lower temperature while in the molten state (92 °С). Nonetheless, there has been a revival of interest in DNA compounds recently because of the development of new DNA salts that exhibit a reduced hygroscopicity and an enhanced thermal stability, inclusive of coordination compounds [14,15,16].

We have previously reported the synthesis of tris(carbohydrazide-N,O)nickel(II)dinitramide by reacting the nickel salt of dinitramic acid with carbohydrazide (CHZ) [17]. This synthesized complex is a nitrogen-rich compound (46%) that is well-soluble in water and insoluble in alcohol, explodes when heated above 180 °С, is very sensitive to impact (the lower limit < 50 mm) and friction (the lower limit = 2200 kgf/cm^2^), and, therefore, is dangerous to work with. However, when tested as a burning-rate modifier, it showed very good positive results towards the inhibition of the burning rate of pyrotechnic compositions, which is typical of high-nitrogen compounds [18,19]. The modifier is employed to adjust and control the burning rate of various energy-rich compositions (gun propellants, СРТТ, pyrotechnic compositions) in order to achieve the required parameters without the need for total re-formulation, which is essential in some cases for pyrotechnic compositions and solid propellants to burn stably [20,21]. Additionally, effective inhibitors can act as flame-extinguishing agents, which are important for fire- fighting in coal mines. Thus, good prospects are seen for the use of tris(carbohydrazide-N,O)nickel(II)dinitramide as a burning-rate retarder, provided that this substance will be insensitive to mechanical stimuli. In this regard, the development of phlegmatizers for tris(carbohydrazide-N,O)nickel(II)dinitramide is highly relevant today. The assortment of such compounds is limited, which is, first of all, due to the stability and chemical compatibility of the dinitramide (DNA) anion with ingredients of a composition. The literature has information on new compounds that can be of potential interest, especially N-heterocyclic macrocycles (NMC, 1,2,4,5,8,9,11,12-octaazacyclotetradeca-5,7,12,14-tetraene-3,10-dione) which hold both the cation and the anion at a time [22].

In this context, using the general synthetic protocol for compound **3** by reacting carbohydrazide with glyoxal in the presence of mineral salts (Figure 1), we were able to synthesize a compound (**3**) bearing X = (N(NO_2_)_2_)^−^, where the nickel salt of dinitramide (**4**) (Figure 1) can be utilized as MeX_2_. It was expected from that synthesis that [Ni(N(NO_2_)_2_)_2_] would react with macrocycle **3**, forming a coordination compound but still remaining as an individual chemical entity―some sort of “encapsulated” into NMC―whereby its physicochemical and detonation performance must improve towards a reduced sensitivity to mechanical impacts and enhanced resistance of the DNA anion to external stimuli, which will allow this substance to be used as a burning-rate modifier in pyrotechnic compositions. The other options we also considered were the use of tris(carbohydrazide-N,O)nickel(II)dinitramide (**5**) as the source that structurally contains CHZ (**1**), and of [Ni(N(NO_2_)_2_)_2_] for the preparation of macrocycle **3** (Figure 2) by addition of glyoxal **2** to **5**.

## 2. Materials and Methods

### 2.1. Materials and Physical Measurements

#### 2.1.1. General Information

UV absorption spectra were taken in water using quartz cells (l = 1.0) at 20 °С by a Varian Cary 50 UV/Vis instrument (Agilent Technologies, Santa Clara, CA, USA). IR spectroscopy of the samples was performed in KBr by a FT-801 Fourier spectrometer (OOO NPF Lumex Sibir, Novosibirsk, Russia) in the range from 4000 to 500 cm^−1^. Elemental analysis was carried out using a CHNO FlashEATM 1112 analyzer (ThermoFisher Scientific, Waltham, MA, USA). A Böetius PHMK (Veb Analytik, Dresden, Germany) instrument was used to determine the melting point. The temperature of decomposition was determined by TGA/SDTA 851e and DSC 822e thermal analyzers (Mettler Toledo, Zürich, Switzerland) in temperature ranges between 25 and 300 °С and between 25 and 500 °С in nitrogen atmosphere, at a heating rate of 10 °С/min. STARe 11.0 thermal analysis software was employed to digitize and process the data. The quantification of Ni was performed on an ICAP 630DVO inductively coupled plasma atomic emission spectrometer. The density was measured on an AccuPyc II 1340 gas displacement pycnometry system (Norcross, GE, USA). The lower limit of friction sensitivity was measured as per GOST R 50835-95. The lower limit of impact sensitivity was measured as per GOST R 4545-88.

#### 2.1.2. Syntheses

Syntheses of [Ni(N(NO_2_)_2_)_2_]*6H_2_O (**4**) and [Ni(H_2_NHNCONHNH_2_)_3_(N(NO_2_)_2_)_2_] (**5**) were carried out according to the procedure [4,17].

##### Synthesis of Adduct **7** from **1**, **2** and **4**

Compound [Ni(N(NO_2_)_2_)_2_]*6H_2_O (4) (1.9 g, 0.005 mol) was dissolved in water (50 mL), and 40% aqueous glyoxal **2** (2.27 mL, 0.02 mol) and carbohydrazide **1** (1.8 g, 0.02 mol) were added with stirring. The whole was held for 4 h at room temperature. The yellow precipitate was collected by filtration, washed with water, and vacuum-dried at 60 °С for 24 h. Yield: 1.60 g (60.0% calculated as **2**). Density: 1.662 g/cm^2^. FTIR (KBr, cm^−1^): 3442, 3213, 1693 (C=O), 1520 (N(NO_2_)_2_), 1363, 1268, 1186 (N(NO_2_)_2_), 1120, 1018, 929, 827, 752. Found (%): С, 22.4; Н, 3.68; N, 38.37; Ni, 4.66. Calculated (%): С, 21.97; Н, 3.66; N, 40.58; O, 29.29; Ni 4.50. Calcd. for С_24_H_48_N_38_NiO_24_. [Ni(N(NO_2_)_2_)_2_]*4(NMC)*8H_2_O. UV/Vis spectrum: the filtrate (78.64 g, 80 mL), the concentration of Ni^2+^ in the solution is 0.28% (0.22 g) (*λ* = 720 nm, *ε* = 2.148, *D* = 0.103), while the concentration of (N(NO_2_)_2_)^-^ is 0.48% (*λ* = 285 nm, *ε* = 5670, *D* = 2.539, V_2_/V_3_ = 1/100).

##### Synthesis of Adduct **8** from **5** and **2**

Compound [Ni(CHZ)_3_(N(NO_2_)_2_)_2_] (5) (1.0 g, 0.002 mol) was dissolved in water (50 mL), and 40% aqueous glyoxal **2** (0.63 mL, 0.00552 mol) was poured. The reaction mixture was held at room temperature for 1 h. The yellow sediment was filtered off, washed with water, and dried in vacuo at 60 °С for 24 h. Yield: 0.82 g (94.7% calculated as **2**). FTIR (KBr, cm^−1^): 3226, 3233, 1693 (C=O), 1523 (N(NO_2_)_2_), 1444, 1367, 1265, 1188 (N(NO_2_)_2_), 1121, 1019, 930, 870, 823, 741, 633. Found (%): С, 23.51; Н, 3.65; N, 39.36; Ni, 3.71. Calculated (%): С, 22.92; Н, 3.82; N, 41.00; Ni, 3.74; O, 31.31. Calcd. for С_30_H_60_N_46_NiO_28_. [Ni(N(NO_2_)_2_)_2_]*5(NMC)*10H_2_O. UV/Vis spectrum, the filtrate (22.93 g, 23 mL) contains: Ni^2+^ (720 nm), (N(NO_2_)_2_)^−^ (285 nm).

##### Hydrolysis of Adduct **7**

Adduct **7** (1.6 g) was added to 70% H_2_SO_4_ (100 mL). The suspension was held with constant stirring for 12 h at 90−95 °С until the sediment was fully dissolved. UV-Vis spectrum, Ni^2+^ (С = 10^−1^ mol/L, *λ* = 720 nm, *ε* = 2.148). Found (%): Ni, 4.51. Calculated (%): Ni 4.50. (N(NO_2_)_2_)^−^ (С = 10^−4^ mol/L, *λ* = 285 nm, *ε* = 5670). Found (g): 0.52. Calculated (g): 0.52.

#### 2.1.3. Pyrotechnic Formulations

(1) 70% potassium perchlorate (chemically pure grade, technical specifications No. 6-09-3801-76, particle size 63−160 µm) and 30% aluminum (ASD-4 grade, technical specifications No. 48-5-226-87, particle size 4−10 µm);

(2) 52% zirconium (PCZr-1 grade (powdered calciothermic Zr), technical specifications No. 48-4-234-84, particle size 1−15 µm) and 48% potassium nitrate (chemically pure grade, GOST R 4217-77, particle size 63−160 µm);

(3) Burning-rate modifier: adduct **8**.

Preparation of ingredients. All the ingredients were dried at 100 °С to constant weight. The oxidizers were ground in a ball mill and sieved to isolate a fraction of 63 to 160 µm.

Preparation of pyrotechnic compositions. The weighed portions of the ingredients of a pyrotechnic composition were taken with an accuracy of 0.0001 g. The ingredients were mixed in an agate mortar.

Burning-rate measurement of pyrotechnic compositions. The tests were performed using cylinder-shaped specimens 10 mm wide, extruded with a force of 20 kg/mm^2^. The burning front passage was recorded on an AKTAKOM ASK-3107 oscillograph with a sampling frequency of 5 kHz.

## 3. Results and Discussion

### 3.1. Synthesis

The objective of the present study was to devise a synthetic method for macrocyclic compounds through the reaction between carbohydrazide and glyoxal in the presence of nickel dinitramide, as well to test the resultant macrocycle as a new burning rate retarder in the burning of some pyrotechnic compositions. The objective set herein is achievable by two ways (Figure 2): (1) a reaction between carbohydrazide **1** and glyoxal **2** in the presence of nickel dinitramide **4**, and (2) a counter synthesis using nickel complex **5** obtained beforehand. 

The methodology of the process for macrocycle **3** via direction (1) (Figure 2) involves the preparation of separate aqueous solutions of compounds **1**, **4**, and **2** and mixing thereof in a specified sequence, in which case a multiple-choice combination is also possible. Direction (2) in Figure 2 involves only the mixing of aqueous solutions of compounds **5** and **2**, where **2** is added to **5**. Compound **2** was expected to react with carbohydrazide of complex **5** to furnish macrocycle **3**. The resultant products obtained via the two directions represent powders that are insoluble in water and organic solvents. For that reason, we faced difficulties (complexities) in purifying and recrystallizing the compound and growing the monocrystal for X-ray diffraction. We failed to take the powder X-ray difraction pattern, as the resultant sediment particles had an amorphous surface and no crystallinity. In view of this, the substance identification was performed by nondestructive techniques, namely infrared spectroscopy and estimations of elemental analysis for C,H,N and atomic emission analysis for Ni. It was anticipated that the same product **3** would be formed in both cases (Figure 2), but the identification results of the sediments obtained via reaction (1) and reaction (2) demonstrated the sediments to be distinct. The gross formula calculation based on elemental analysis and Ni cation content showed that the first case afforded four macrocycles of **6**, which is obtained by reacting **1** with **2** [23], for one [Ni(N(NO_2_)_2_)_2_] molecule, whereas the second case afforded **5** macrocycles of **6** for one [Ni(N(NO_2_)_2_)_2_] molecule, given that there were two H_2_O molecules for each macrocycle. Thus, the resultant reaction products can be referred to the adducts and the scheme of producing the same (**7** and **8**) can be depicted as below (Figure 3).

Alternatively, this can be represented as mixed products of macrocycle **3** with monocycle **6** (Figure 4).

If it were mixed products **3** and **6**, the IR spectra would have been identical and no difference observed.

A comparative analysis of the IR spectra of the resultant precipitates **7** and **8** (Appendix A) showed them to have absorption bands of the functional groups belonging both to methyl-free macrocycle 6 synthesized by the reported procedure [23] and to compounds **4** and **5** (Table 1). It is seen from Table 1 that the absorption band of characteristic stretching vibrations of the NO_2_ groups is correlated very well, that is, (*ν* assym.) 1537 cm^−1^ (5) > 1528 cm^−1^ (4) > 1523 cm^−1^ (7) > 1520 cm^−1^ (8) and (*ν* sym.) 1188 cm^−1^ (8) > 1187 cm^−1^ (7) > 1180 cm^−1^ (4) > 1177 cm^−1^ (5), suggestive of intermolecular and intramolecular interactions between the DNA nitro group and the functional groups of adducts **7** and **8**. The absorption bands near 1539 cm^−1^ and 1455 cm^−1^ are inherent in compound **1** and are absent in **7** and **8**, indicative of a complete interaction between compound **1** and glyoxal **2**.

The estimation of the structure from elemental analysis and Ni content demonstrates that there are four to five molecules of macrocycle **6** for one molecule of [Ni(N(NO_2_)_2_)_2_], with two molecules of water for each macrocycle 6, which is in agreement with the data from [22] and performed TGA analysis, as well as with the weight losses from 8.8% to 9.4% at the first stage and from 12.6% to 20.4% at the second stage of decomposition of **7** and **8**.

The DSC and TGA analyses (Appendix A) demonstrated that compounds **7** and **8** have three decomposition stages (Table 2): for instance, the first stage (decomposition of the (N(NO_2_)_2_)^−^ anion) exhibits a decomposition peak between 186 °С and 198 °С with a specific heat release of 246 to 268 J/g in the DSC curve; at the second stage (decomposition of macrocycle **3**), the intense decomposition peak is between 270 °С and 273 °С with a specific heat release of 396 to 436 J/g; and at the third stage (synthesis products from the decay at the first and second stages), the intense decomposition peak is between 419 °С and 443 °С with a specific heat release of 2198 to 2993 J/g. For comparison, the DSC of compound **3** reported in [23] showed only one decomposition stage at 298 °С with a specific heat release of 1056 J/g in the DSC curve, while complex **5** has only one decomposition peak at 217 °С.

The resultant decomposition temperature intervals and specific heat release intervals between experiments can be explained by compounds **7** and **8** having different contents of compound **4**, depending on the synthetic method. The studies demonstrate that compounds **7** and **8** structurally contain energetic components like the DNA anion that enhance the heat release on the decomposition of the substance, and the (Ni^2+^) components that elevate the decomposition temperature. The complexes are fairly stable in the air. The hydrated [(NMC) (H_2_O)_2_] complexes gradually lose water when heated above 50°С. The decomposition of NMC in **8** can be represented by the following scheme based on the TGA results (Figure 5):

An attempt to derive compound **3** by the scheme shown in Figure 4 through the prolonged refluxing of compound **6**, obtained by the procedure reported in [22], in an aqueous solution of **4** did not lead to the anticipated result. By cooling the aqueous solution, we detected pretty crystals which were standalone from the amorphous sediment (starting compound **3**) and were identified by IR spectroscopy, as was compound **9** (Figure 6) that was confirmed by X-ray diffraction. This compound had previously been identified by X-ray diffraction [24].

Adduct **7**, when thermally treated (90−95 °С) with dilute (60−70%) mineral acids (H_2_SO_4_, HNO_3_), degraded to liberate compound **4** which was identified in the solution by UV-Vis spectroscopy, by the presence of Ni^2+^ (*λ* = 720 nm) and (N(NO_2_)_2_)^−^ anion (*λ* = 285 nm) and by comparing the IR spectrum of the isolated (KN(NO_2_)_2_) sediment, resulting from the neutralization of compound **4** with alkali (KOH) and evaporation of the acetone extract, with the IR spectrum reported in the literature [4].

Thus, a set of studies done demonstrate that adducts **7** and **8** obtained from compounds **1**, **2**, **4**, and **5** in two directions by different combinations are almost identical and contain the (Ni^2+^) cation, (N(NO_2_)_2_)^−^ anion and hydrated macrocycle **3**, which all can be expressed by the general formula: [Ni(N(NO_2_)_2_)_2_]*(NMC)_n_*2(H_2_O)_n_, where n = 4−5.

### 3.2. Burning-Rate Measurements

To test how compound **8** influences the burning process of pyrotechnic compositions extruded into specimens, we chose two low-gas compositions to preclude the effect of resulting gases within the chemical action zone. The compositions differed in burning rate and were made from the most common components used in pyrotechnics. The formulations were selected such that the potential action of both catalytic and inhibitory additives could be assessed (Appendix A).

This way, the first model composition was a formulation containing 70% KClO_4_ and 30% Al whose burning rate is not high: the burning rate of this formulation without additives is 3.8578 mm/s. The second model composition was a formulation comprising 58% Zr and 42% KNO_3_. This formulation is characterized by a higher burning rate of 34.6617 mm/s. 

To explore the role of the complex salts, they were incorporated into the chosen formulations in small quantities of 0.5%, 1.0% and 1.5% on top of 100% of the weight.

Initially, we evaluated the effect of modifier **8** on the burning rate. The results from evaluation of the impact of modifier **8** on the burning rate of the pyrotechnic compositions are illustrated in Figure 7 and Figure 8.

The data on the change in the burning rate of KClO_4_/Al when **8** was added (Figure 1) demonstrate a decline in the burning rate as the modifier content in the formulation was raised. This way, the burning rate was observed to decline smoothly from 3.8578 mm/s to 2.3835 mm/s. The data on the change in the burning rate of KClO_4_/Al when **8** was added (Figure 7) shows that the burning rate was reduced by increasing the modifier content in the pyrotechnic composition. This way, the burning rate was observed to decline smoothly from 3.8578 mm/s to 2.3835 mm/s.

The results from evaluation of the impact of added modifier **8** on the burning rate of the KNO_3_/Zr composition demonstrate that the burning rate of this system was reduced from 33.9783 mm/s to 20.0348 mm/s by incorporating and further increasing the content of modifier **7** from 0.5% to 1.5% in the composition (Figure 8).

Table 3 summarizes the data on the burning rate decrease levels when compound **8** was used as the burning-rate retarder.

It is seen from Table 3 that the inhibiting ability of compound **8** from is 38.2% when its concentration is 0.5% to 1.0% in the KClO_4_/Al formulation, and is as high as 41% when its concentration is 0.5% to 1.5% in the KNO_3_/Zr composition.

The density of compound **7** as measured by the helium pycnometer was 1.662 g/cm^3^.

Among the most essential properties of energetic materials is the sensitivity to mechanical stimuli; thus, the lower impact sensitivity limit of **7** was 120 mm at a drop weight of 10 kg, while the lower friction sensitivity limit was 5500 kgf/cm^2^ (539.36 MPa), which is almost two times less sensitive than compound **5** [17].

## 4. Conclusions

1. One-pot methods have been devised for the “encapsulation” of the nickel salt of dinitramic acid into a macrocycle (1,2,4,5,8,9,11,12-octaazacyclotetradeca-5,7,12,14-tetraene-3,10-dione) by reacting carbohydrazide with glyoxal.

2. The resultant macrocycle bearing nickel dinitramide improves the performance of the latter because, first of all, nickel dinitramide is no longer soluble in water and organic solvents, has no hygroscopicity peculiar to the dinitramide salts, and has a reduced sensitivity to impact and friction.

3. The new metal-containing macrocyclic complex salt has been tested as a burning-rate modifier of pyrotechnic compositions. The nickel dinitramide “encapsulated” into the macrocycle was found to lower the burning rate of the KClO_4_/Al and KNO_3_/Zr pyrotechnic compositions, suggestive of a potential use of this macrocycle as a burning-rate retarder of pyrotechnic compositions when fabricating extruded pyrotechnic grains.

## Figures and Tables

**Figure 1 materials-15-06958-f001:**
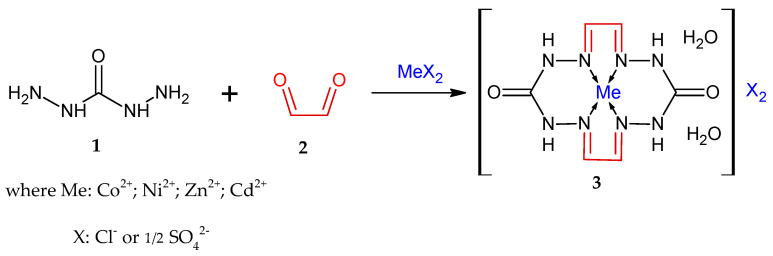
Synthesis of macrocycle (**3**) by reaction with glyoxal.

**Figure 2 materials-15-06958-f002:**
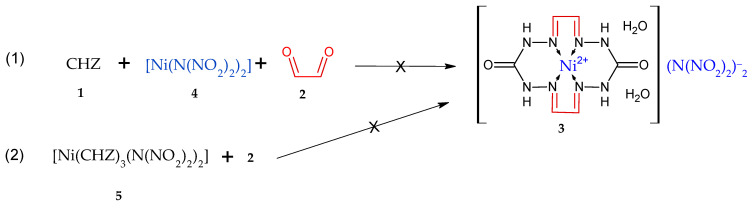
A synthetic scheme for macrocyclic complex **3** in two directions.

**Figure 3 materials-15-06958-f003:**
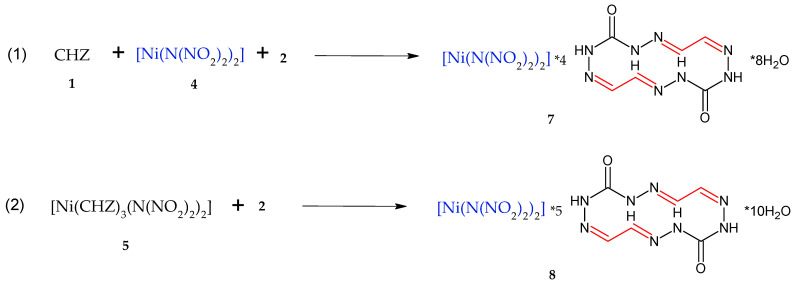
Synthesis of adducts **7** and **8**.

**Figure 4 materials-15-06958-f004:**
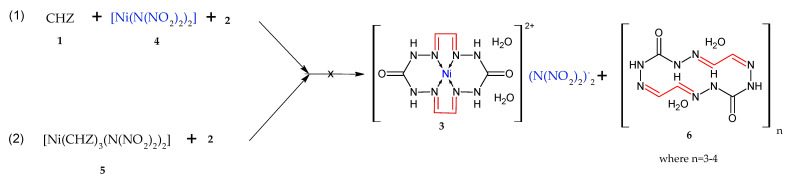
Mixed products **3** and **6**.

**Figure 5 materials-15-06958-f005:**
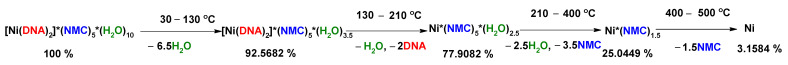
A block diagram of thermal decay of **8**.

**Figure 6 materials-15-06958-f006:**
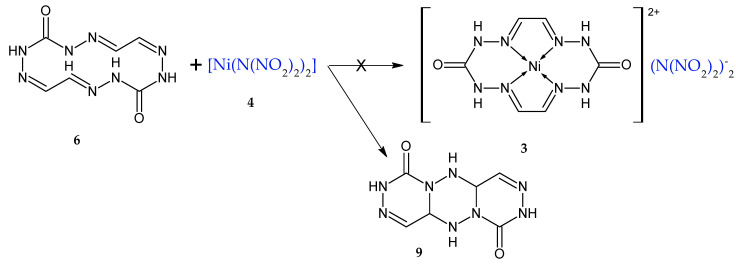
A synthetic scheme for **9** from **3**.

**Figure 7 materials-15-06958-f007:**
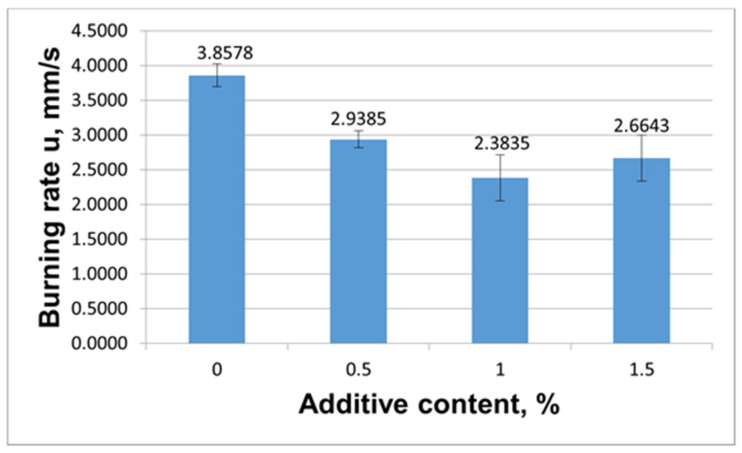
Effect of the content of modifier **8** on the burning rate of the KClO_4_/Al.

**Figure 8 materials-15-06958-f008:**
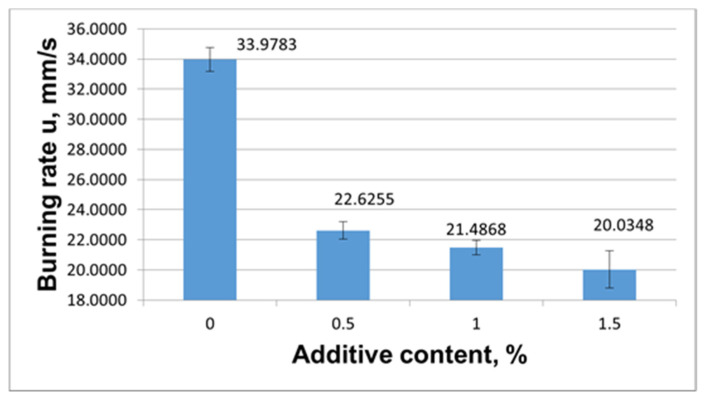
Effect of modifier **8** on the change in the burning rate of KNO_3_/Zr.

**Table 1 materials-15-06958-t001:** Comparative basic IR frequencies of **4**, **5**, **6**, **7**, and **8**.

Entry	Frequency, cm^−1^						
**7**	1693 v.s.	1605 w.	1520 v.s.	1268 m.	1187 s.	1018 w.	827, 752
**8**	1693 v.s.	1605 w.	1523 v.s.	1265 m.	1188 m.	1019 w.	823
**6** [23]	1691 s.	1514 s.	Abs.	1265 s.	Abs.	1008 w.	Abs.
**4**	Abs.	Abs.	1528 v.s.	Abs.	1180 v.s.	1023 s.	828, 762, 731
**5** [17]	1644 v.s.	1604 w., 1542 с,	1537 v.s.	1283 m	1202 v.s., 1177 v.s.	1022 s.	828, 762, 732
Functional groups	*ν* C=O	*δ* NH, *ν* C=N	*ν* NO_2_	*ν* C-N	*ν* N=N-O	N-N N-N-N	NO_2_

IR intensities: s., strong; m., medium; w., weak; v., very; sh., shoulder; Abs., absent; ν, stretching; δ, bending. Font color: red, the prevailing absorption band of compound 4; blue, the prevailing absorption band of compound 3.

**Table 2 materials-15-06958-t002:** DSC data on decomposition temperatures of **1**, **5**, **6**, **7** and **8**.

Entry	First Stage				Second Stage				Third Stage			
	Onset, °С	Peak, °С	Endset, °С	Specific Heat Release, J/g	On Set, °С	Peak, °С	Endset, °С	Specific Heat Release, J/g	On Set, °С	Peak, °С	Endset, °С	Specific Heat Release, J/g
**1**	**2**	**3**	**4**	**5**	**6**	**7**	**8**	**9**	**10**	**11**	**12**	**13**
**1**	156	157 *	160	−321	-	-	-	-	-	-	-	-
**6** [23]	-	-	-	-	288	298	291	1056	-	-	-	-
**5** [17]	214	217	218	1619	-	-	-	-	-	-	-	-
**7**	160	186	216	212	246	270	288	396	330	419	423	2993
**8**	175	198	234	154	268	273	275	599	436	443	446	2198

* Melting point.

**Table 3 materials-15-06958-t003:** Burning rate as a function of concentration of modifier **8** in the formulation.

Additive	∇u (KClO_4_/Al), %	∇u (KNO_3_/Zr), %
0.5% **8**	−23.82	−33.41
1.0% **8**	−38.21	−36.76
1.5% **8**	−30.94	−41.03

## Data Availability

Not applicable.

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
