# Peer review of "Macrocycle as a “Container” for Dinitramide Salts"

_materials, 2022, doi:10.3390/ma15196958_

Round 1

Reviewer 1 Report

The work is about the preparation of the adduct of the nickel salt of dinitramic acid with carbohydrazide and glyoxal. It is a well-designed experiment and manuscript. Some issues are remaining to be addressed to improve the quality of the manuscript. 

(1) Structure analysis of the compound by only the use of FTIR and elemental analysis is not enough. Since the product is solid, it could be better if more analyses by XRD. 

(2) Based on DSC and TGA data, the detailed reaction during heating (Fig 3) could be better explained to understand the detail of each stage of weight loss (the produced gases and residue). Is the final residue metallic Ni? 

(3) It seems the reference is not enough. More and updated references are required.

(4) Table 1 was not well organized, it could be simplified by only focusing on the specific functional groups. 

(5) Ref 1 was not properly written.

(6) The writing of 'degree' Celcius is not appropriate. Check the common and accepted symbol/unit.

(7) The order of writing of the pyrotechnic formula should be consistent, salt/metal?

(8) The assignment of the synthesized compounds is better by the simple number and consistency. For example, replace 3_1 with other. Likewise for the experimental such as exp. 2.2.2.2, it may be simplified.    

Author Response

The response to the reviewer's comments has been uploaded as a separate PDF file.

Reviewer 2 Report

I originally intended to give this work a chance to continue to revise. However, I can only say that the author of this manuscript did not take it seriously in the preparation process. Including abstracts, discussions and references.Some necessary academic norms are recommended to be followed. For example, why not use Origin lab for the presentation of tables  and Figures?I see more discussion on factors than in-depth mechanism analysis.Therefore, I should rejected this manuscript at this stage. I hope the author can seriously and carefully improve the quality of the manuscript.

Author Response

(The authors gave the same response as above.)

Reviewer 3 Report

The paper is devoted to the synthesis macrocycle (NMC) that contains a complex compound based on nickel (II). 

The introduction was not shown a very well the novelty and the application. There was a list of a few references that does not show the most relevant research.    

Author Response

(The authors gave the same response as above.)

Reviewer 4 Report

This manuscript reports a study of dinitramide salt. Overall, this report presents a comprehensive work. However, this reviewer felt that the importance of this work was not sufficiently highlighted. This has somewhat deteriorated the technical merit of this work. Some specific comments are provided below for authors’ consideration to improve this manuscript before its publication.

1. Use a structured abstract. It is important to highlight your results, but plz also address the advance and importance of your work briefly.

2. it is useful for you and your co-authors to provide that background (with references) and then explain what is new and novel about your work.  

3. Plz provide more references to provide sufficient background information, highlight your approach, and address the importance of your work.

4. plz pay attention to details. E.g., 270 °C instead of 270°C, λ (italic) instead of λ

Author Response

(The authors gave the same response as above.)

Round 2

Reviewer 2 Report

The content of the article is acceptable. But only the content. However, I wonder why there are so many details in this latest version of the manuscript that have not been corrected? Does the author take his work too seriously? For example, in the references, there are many detailed errors, some in italics and some in non italics. Does the author not check when he returns the review comments? The material is a very good journal, and this basic mistake should be avoided. In addition, I have suggested that you draw with professional origin instead of the less professional presentation of cancel. A large number of literatures have gradually adopted this mapping method. Why does the author dislike correction? There are also significant errors in the chemical formula of experimental materials, especially drugs. The author should carefully check and correct it.

Author Response

The author's response to Reviewer 2 has been uploaded as a separate file.

Reviewer 3 Report

The authors accepted the comments and made revisions.

Therefore this manuscript can be accepted for publication.

Author Response

The author's response to Reviewer 3 has been uploaded as a separate file.
